# Enhanced Pitting Corrosion Resistance of Nanostructured AISI 304 Stainless Steel via Pipe Inner Surface Grinding Treatment

**DOI:** 10.3390/nano13020318

**Published:** 2023-01-12

**Authors:** Xiaolei Han, Ping Wei, Yiming Zhao, Zuohua Wang, Changji Li, Xinqiang Wu, Hongwang Zhang

**Affiliations:** 1National Engineering Research Center for Equipment and Technology of Cold Strip Rolling, College of Mechanical Engineering, Yanshan University, Qinhuangdao 066004, China; 2State Key Laboratory for Advanced Metals and Materials, University of Science and Technology Beijing, Beijing 100083, China; 3College of Weaponry Engineering, Naval University of Engineering, Wuhan 430033, China; 4Shenyang National Laboratory for Materials Science, Institute of Metal Research, Chinese Academy of Sciences, Shenyang 110016, China

**Keywords:** pitting corrosion, nanostructure, AISI 304 stainless steel, pipe inner-surface grinding

## Abstract

By means of a pipe’s inner surface grinding, a single-phase nanostructured austenite was formed on the surface of an AISI 304 stainless steel. The electrochemical corrosion behavior was compared with a coarse-grained counterpart of identical surface roughness. Experimental results show that the nanostructured austenite shows a higher pitting potential and a wider passivation interval than those of its coarse-grained counterpart. The enhanced corrosion resistance was attributed to the fast diffusion of Cr within the nanostructure and, hence, the formation of a thicker passive film to efficiently protect the surface against the ion attack. This work provides insights into a simple processing method to improve the surface strength and pitting resistance of stainless steel.

## 1. Introduction

In the past decades, great attention has been paid to the nanostructuring material surface, by which the surface properties as well as the global service performance can be enhanced without altering the chemical composition [1,2]. The relationship between grain refinement and the surface properties has been of great concern and, hence, was extensively investigated, among which corrosion is of great importance, in particular for materials serviced in a corrosive environment. However, because corrosion is very complicated and is influenced by many factors, such as temperature, pH value, surface quality, phase constitution, etc., the correlation of nanostructure with corrosion resistance from different publications varies and sometimes is even contradictory [3]. It is reported that nanostructured surfaces of AISI 316 stainless steel prepared by surface mechanical attrition treatment (SMAT), namely, high energy shot bombardment on the sample surface, exhibit decreased corrosion resistance in acidic media containing SO_4_^2−^ [4]. On the contrary, nanostructuring the surface via the shot peening process improves the corrosion resistance of a Ni-22Cr-13Mo-4W alloy [5]. For an AISI 304 stainless steel, the grain refinement reduces the general uniform corrosion performance and intergranular corrosion resistance but improves the pitting corrosion resistance [6]. It is also reported that corrosion performance is related to the electrolyte environment. The nanostructured surface of the FeAl alloy in the active electrolyte shows lower corrosion performance, but in the electrolyte that promotes passiveness, it is more resistant [7]. Consequently, addressing the intrinsic relationship between nanostructure and the corrosion behavior under a given corrosion environment is usually influenced by surface roughness and the extra phase, in particular for metastable austenitic steels that are susceptible to deteriorating surface quality and the martensitic transformation during nanostructuring via surface plastic deformation [8].

Given previous work, if the key factors in the corrosion process of nanostructured metals can be controlled, it is desirable to study the internal influence of nanostructures on corrosion behavior. As to metastable austenitic stainless steel, the key factors include surface roughness, especially second phase particles. The surface roughness can be controlled by precise machining [2,9], but unfortunately, many previous processing methods [3,4,6,8] are susceptible to inducing phase transformation of metastable austenitic stainless steel and producing the extra phase, which has not been optimized; there is room to further enhance.

Recently, we developed a surface nanostructuring technique, i.e., pipe-inner surface grinding (PISG), that can produce a very thick nanostructured layer [10,11]. More interestingly, the high strain rate shear as well as the adiabatic heating suppress the deformation induced phase transformation. It is demonstrated that the surface of the PISG-processed metastable austenitic AISI 304 stainless steel is composed of single-phase nanostructured austenite of ~80 μm thickness [11]. This provides an opportunity to reveal the influence of grain refinement on pitting corrosion resistance for AISI 304 stainless steel without the influence of surface roughness and extra phases, which sheds light on a new way to investigate the corrosion behavior of nanostructures. Consequently, the pitting resistance of a nanostructured surface in a 0.6 mol/L NaCl solution was compared with that of its coarse-grained counterparts. The enhanced corrosion resistance by nanostructuring was underpinned, and the mechanism was analyzed based on the detailed characterization of the passive film.

## 2. Materials and Methods

The material used in this investigation is an AISI 304 stainless steel pipe (80 mm in inner diameter and 110 mm in outer diameter) with chemical compositions (in wt.%): 0.049C, 18.46Cr, 8.28Ni, 0.012M0, 1.64 Mn, 0.42Si, 0.021P, 0.03S, and a balance of Fe. The material for PISG processing and the controlled experiment were subjected to solution treatment at 1080 °C for 1 h, followed by cooling into water, so as to produce a single austenite phase free of residual stress. PISG produces nanocrystallites through high strain rate shear deformation via ball tips that rotate around (*V*_1_) and concurrently move (*V*_2_) along the pipe axis after penetrating the surface for a depth (*D*) [10]. As the ball tips moved from one end to the other, the sample was processed in one pass. The above operation repeats so as to introduce multipass processing, during which the temperature rise was controlled by using cooling water. Here, *V*_1_ = 80 rpm, *V*_2_ = 50 mm/min, and *D* = 100 μm, and the materials were subjected to 4-pass processing.

Electrochemical experiments were performed using an EG&G 273A electrochemical work station. Slices of 15 × 15 × 2.5 mm were cut from the starting and PISGed samples. Then, the sample surface was mechanically polished with sandpaper followed by electrochemical polishing (electrolyte of 10 vol.% HClO_4_ and 90 vol.% C_2_H_5_OH) to remove the affecting layers and produce similar roughness. The samples were sealed with epoxy resin, leaving a testing surface of 10 mm^2^ exposed for the pit corrosion testing. Corrosion properties were evaluated by potentiodynamic polarization measurements using a three-electrode apparatus in a 0.6 mol/L NaCl aqueous solution at ambient temperatures. During the tests, a saturated calomel electrode (SCE) acted as a reference electrode and was placed between the tested sample (working electrode) and the platinum plate (counter electrode). Potentiodynamic polarization measurements were carried out in the potential range from −250 mV (SCE) in the cathodic direction to +800 mV (SCE) in the anodic direction from open circuit potential (OCP) at a scan rate of 1.67 mV/s. Before the electrochemical tests, samples were stabilized in an NaCl solution for 30 min.

X-ray diffraction (XRD) (Rigaku, Tokyo, Japan) was used to determine the phase constitution in the surface of the staring and the PISG samples with a Rigaku D/max-2400 X-ray diffractometer with Cu target. The microstructure of the PISG samples was detailed by optical microscopy (OM), FEI-Scios scanning electron microscopy (SEM), and FEI-Talos 200X transmission electron microscopy (TEM, operated at 200 kV) (Thermo Fisher Scientific, Waltham, MA, USA). The element distribution within the passive film was characterized by energy disperse spectroscopy (EDS) (Thermo Fisher Scientific, Waltham, MA, USA) on the TEM.

## 3. Results

In Figure 1a, the starting coarse-grained sample is composed of equiaxed grains with clean interiors and frequent annealing twins having straight and parallel boundaries. The coarse-grained sample has a grain size range of 10–200 μm, with an average of 150 μm. After PISG, nanocrystallites layer of ~80 μm thick was produced, followed by high deformed layer composed of mainly shear bands, deformation twins and high density of dislocations [10]. After a surface layer 20 μm thick was removed from the position indicated by the dashed line, the sample was subjected to careful mechanical and electrochemical polishing to obtain the identical roughness (*Ra* = 0.49 μm) as that (*Ra* = 0.47 μm) for coarse-grained sample. The XRD examination (Figure 1c) shows that only the single austenite phase with face-centered cubic (fcc) lattice structure was detected in the nanocrystalline sample, in accordance with that of the coarse-grained sample. Comparatively, each diffraction of the nanocrystalline sample is apparently broadened, implying significant structural refinement and the enhancement of crystal defect density. As demonstrated by the TEM observation (Figure 1d), a nanocrystalline sample is composed of a nanoscale lamellar structure with the boundaries roughly parallel to the shear direction. Within lamellae interiors, interconnecting boundaries and loose dislocations are presented. The selected area electron diffraction pattern shown in Figure 1e further shows the existence of a single-phase austenite (γ). The spacing between lamellar boundaries ranges from 5 to 140 nm, with an average of 45.6 nm, Figure 1f.

The pit corrosion behaviors of nanograined and coarse-grained samples were examined in terms of potentiodynamic anodic polarization curves in aqueous solutions of 0.6 mol/L NaCl without deoxygenation. In Figure 2, both nanograined and coarse-grained samples exhibit typical passive behavior, where current density *I* changes within wide passive ranges. In the passivation interval, the passive current density increases with an increase in potential for both coarse-grained and nanograined samples, but the nanograined sample changes relatively slowly. Additionally, the nanograined sample has a higher pitting potential (nanograined ~22 mV and coarse-grained ~−45 mV, respectively) and a wider passive interval (nanograined from ~−201 to 22 mV and coarse-grained from ~−141 to −45 mV, respectively) than the coarse-grained samples, indicating that the PISG enhances the resistance against pit corrosion to some extent.

The passive film formed on the coarse-grained sample was further characterized by an electron microscope. The cross-sectional TEM sample was created by a focused iron beam (FIB) equipped on the SEM. As seen in Figure 3a, Pt was deposited on the surface to protect the passive film from the damage caused by Ga iron bombardment. The coarse-grained sample was composed of coarse grains with a high density of dislocations that might be induced during sample preparation. A high-resolution TEM (HRTEM) image of the white rectangle marked area, Figure 3b, shows that a very thin passive film was formed. As demonstrated by the line scanning by energy dispersive spectroscopy (EDS) along the direction shown in (b), Figure 3c, Cr is enriched within the passive film. This can be further manifested by the map scanning of the rectangle-marked area in (b), Figure 3e,f, where an area of 4 nm thick with slight Cr enrichment was observed. Such an area corresponds to the passive film formed during pit corrosion. A close observation of the HRTEM image of the passive film, Figure 3d, finds that the lattice fringe disappears in some areas but is preserved in others, implying the feature of a mixture of crystal and amorphous structure.

From the TEM characterization of the nanograined sample (Figure 4a), a layer with bright contrast was observed between the nanograined matrix with nanoscale lamellar boundaries and the deposited Pt layer. HRTEM images of this bright layer, Figure 4b,d, reveal the amorphous feature characterized by the absence of the lattice fringe in the outer layer, which is a crystal feature close to the nanograined matrix. EDS line scanning and mapping, Figure 4c,e,f, underpin the thick passive film of 8 nm enriched with Cr but depleted of Fe. As shown in Figure 4b, the nanostructure, the Pt layer, and the thick passive film in the middle can be clearly observed. The HRTEM images of the passive film, Figure 4b,d, are similar to those in [12,13], showing that the passive film of the 304 stainless steel after PISG is mostly amorphous, and shows no obvious orientation relationship with the matrix. Figure 4c shows the EDS line scan of passive film, from which Fe is found to gradually decrease as it moves from the matrix, while the Cr first increases and then decreases. The passive film has the highest chromium content, indicating that Cr is enriched in the passive film, which is further evidenced by the map scan (Figure 4e,f).

## 4. Discussion

Here, the present study demonstrates that AISI 304 stainless steel, as its structural scale was reduced down to nanometer scale without sacrifice the surface quality and the phase composition, the resistance against pit corrosion within 0.6 mol/L NaCl aqueous solution is superior to the coarse-grained counterpart. The corrosion resistance is known to be influenced by many factors, among which the surface quality and the second phase are most relevant to the AISI 304 stainless steel. It is reported that the electrochemical behavior, represented by the electronic work function (EWF, which refers to the minimum energy required to move electrons outside of the solid) becomes worse with an increase in the surface roughness [14]. Theoretically, surface roughness can decrease the average but increase the fluctuation of the local EWF. Such fluctuation is able to promote the formation of microelectrodes and, therefore, accelerate corrosion [14]. Recent studies [15,16] also show that the electrons in the vicinity of the peak are easier to escape than those in the valley, and that the peak may have a higher tendency to corrode than the valley. High roughness leads to a decrease in pitting potential. Furthermore, the difference in EWF between peak and valley may also promote local corrosion and, hence, accelerate global corrosion. A decrease in the amount of second phase is able to enhance the corrosion resistance of fine-grained materials [8]. The machining of stainless steel followed by post-heat treatment was found to enhance corrosion performance owing to the efficient removal of the second phase induced by processing [17,18,19]. Second-phase particles and the matrix can build microcells during the electrical and chemical processes and/or act as initiation sites for localized corrosion [18]. The XRD (Figure 1) and TEM (Figure 3) show that PISG-induced nanostructured single-phase austenite, which is desired for stainless steels against corrosion. The temperature rise induced by high-speed shearing and adiabatic heating plays an important role, under which the driving force for the transformation from austenite to martensite is reduced [10]. The comparable surface roughness, together with the suppression of the second phase, allow the size effect of pit corrosion for AISI 304 stainless steel to be properly addressed.

The potentiodynamic polarization tests in Figure 4 demonstrated that the present nanostructure formed on AISI 304 stainless steel shows enhanced corrosion performance. Not only the pitting potential but also the pitting interval was increased relative to that of the coarse-grained sample in the NaCl solution. By PISG, the nanostructure layer with a mean grain size of approximately 45.6 nm provides a considerable volume fraction of grain boundaries (GBs) and triple junctions (i.e., the intersection pipes of three adjoining crystals). With the decrease in grain size, the volume fraction GB increases drastically [20]. GBs and triple junctions may serve as channels for rapid diffusion of elements to increase the diffusion rate. From a kinetic aspect, the Cr atom is expected to migrate to the sample surface with a significantly increased diffusion rate in the nanostructured surface layer. The effective diffusion coefficient (*D_eff_*) of atoms in a polycrystalline material can be expressed as [21]: *D_eff_* = (1 − *f*) *D_V_* + *fD_B_*, where *f* is the fraction of GBs, *f* = 2*δ*/*d* (*δ* is the GB width), and *D_V_* and *D_B_* are the diffusion coefficients in the crystal lattice and along GBs, respectively. For a nanograined and coarse-grained sample, the grain sizes of 45.6 nm and 150 μm, give rise to *f* of 4.4% and 0.0013%, respectively. The *D_eff_* of Cr in the surface nanograined sample is estimated to be ~2.9 × 10^3^ times higher than that in the coarse-grained sample, which is close to the ratio of *D_B_* to *D_V_* [22,23].

The high corrosion resistance of stainless steel is known to stem from the formation of a Cr-enriched passive film, which protects the matrix from corrosion attack [24]. The highly enhanced Cr diffusivity of the nanograined surface promotes the formation of a thick passive film (Figure 4). Additionally, the passive film formed on the nanograined 304 stainless steel has a stronger protective effect [25]. Here, we demonstrate that the corrosion potential of the nanograined sample is more negative than that of the coarse-grained samples, indicating a stronger corrosion tendency and a faster corrosion rate of the nanograined sample due to the large number of defects. Since the corrosion product is an insoluble, stable oxide with excellent element diffusion ability, it promotes the advent of the passive stage and inhibits further corrosion. There are some reports [4,26] showing that nanostructured surface layers produced by plastic deformation reduce corrosion resistance. These results, together with the present study, underpin the fact that corrosion resistance is influenced by many positive or negative factors, including processing, materials, and surface state. When positive effects (high diffusion rate, single-phase austenite, etc.) exceed the negative ones, corrosion is enhanced; otherwise, the opposite conclusion is drawn [27,28]. The exact combination of environment, processing, and material will determine whether the increased surface reactivity will increase or decrease corrosion resistance [29].

The improved pitting resistance by grain refinement can be attributed to an improved passive film, agreeing with previous investigations [5,30]. Nanostructure influences the performance of the passivation film in different manners and, hence, induces different corrosion behaviors [25,26,27,28,29,30,31]. It is reported that a nanostructured surface increases the volume fraction of GBs and the surface reactivity and provides more sites for the nucleation of an oxide film, leading to a more uniform oxide at a faster speed [30]. It is also found that the change in surface state caused by the nanostructure affects the adsorption capacity of ions, which changes the semiconductor properties of the film, thus, reducing or improving the corrosion resistance of the material. The nanostructure is found to increase the thickness of the passive film in this work. As shown in Figure 3 and Figure 4, the film formed on the surface of the nanostructure is roughly double that of the coarse-grained sample and is approximately 6 nm thicker than the natural oxide film formed on the surface of the Fe-18Cr-13Ni single-crystal sample [31]. The passivation film on the surface of AISI 304 stainless steel is known mainly as the oxide produced by the corrosion reaction, which hinders the dissolution of the material and inhibits corrosion. However, in solutions with aggressive anionic species (such as Cl^−^), the effect of the passive film in inhibiting corrosion is reduced [32]. The nanostructure increases the film thickness because of the increased surface reaction activity, the fast diffusion of Cr and the increased number of reaction products. The thicker film is able to decrease the rate of destroying passive film by Cl^−^, making pitting breakdown more difficult. The breakdown of the passive film is apparently related to the microstructure, element distribution of passive film, as well as the features of the reaction interface, metal/film and film/solution interfaces, etc., which will be analyzed in our forthcoming paper.

## 5. Conclusions

In summary, single-phase nanostructured austenite was formed on the surface of an AISI 304 stainless steel via PISG at room temperature. The pit corrosion behavior in a 0.6 mol/L NaCl solution of the nanostructured austenite (45.6 nm) was studied comparatively with its coarse-grained (150 μm) counterpart with the same surface roughness. The main results of the study have been summarized as follows:Both nanostructured and coarse-grained austenite exhibit passivation behavior. However, the nanograined sample has a higher pitting potential (nanograined ~22 mV and coarse-grained ~−45 mV, respectively) and a wider passive interval (nanograined from ~−201 to 22 mV and coarse-grained from ~−141 to −45 mV, respectively) than the coarse-grained samples, indicating that the nanostructured sample has higher resistance to pitting corrosion.The nanostructured sample forms a passive film of 8 nm enriched with Cr but depleted of Fe, which is thicker than the passive film of 4 nm of coarse-grained samples. The nanostructured sample forms a thicker passive film since Cr diffusion was enhanced. The effective diffusion coefficient (*D_eff_*) of Cr in the surface nanograined sample is estimated to be ~2.9 × 10^3^ times higher than that in the coarse-grained sample_._Here, the positive effect of grain refinement was highlighted, and the side effects on corrosion resistance from surface roughness and the second phase should be deliberately considered.

## Figures and Tables

**Figure 1 nanomaterials-13-00318-f001:**
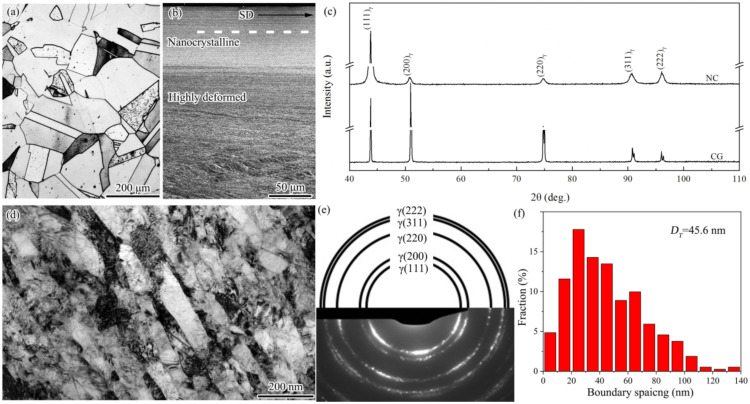
OM image (**a**), cross-sectional SEM observation (**b**), XRD examination (**c**), and TEM characterizations (**d**–**f**) of the starting coarse-grained and PISG-processed nanograined AISI 304 stainless steel samples. Nanograined sample was taken at the surface layer from the position indicated by the dashed line. Shear direction (SD) is marked by a black arrow. Note that single-phase austenite with an average grain size of 150 μm for the coarse-grained sample and 45.6 nm for the nanograined sample will experience controlled pitting corrosion resistance testing.

**Figure 2 nanomaterials-13-00318-f002:**
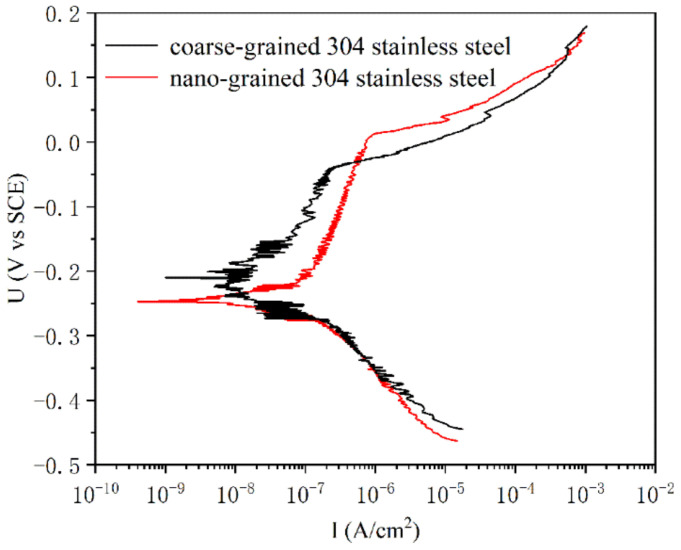
Potentiodynamic anodic polarization curves of nanograined and coarse-grained AISI 304 stainless steel with aqueous solutions of 0.6 mol/L NaCl without deoxygenation at ambient temperatures.

**Figure 3 nanomaterials-13-00318-f003:**
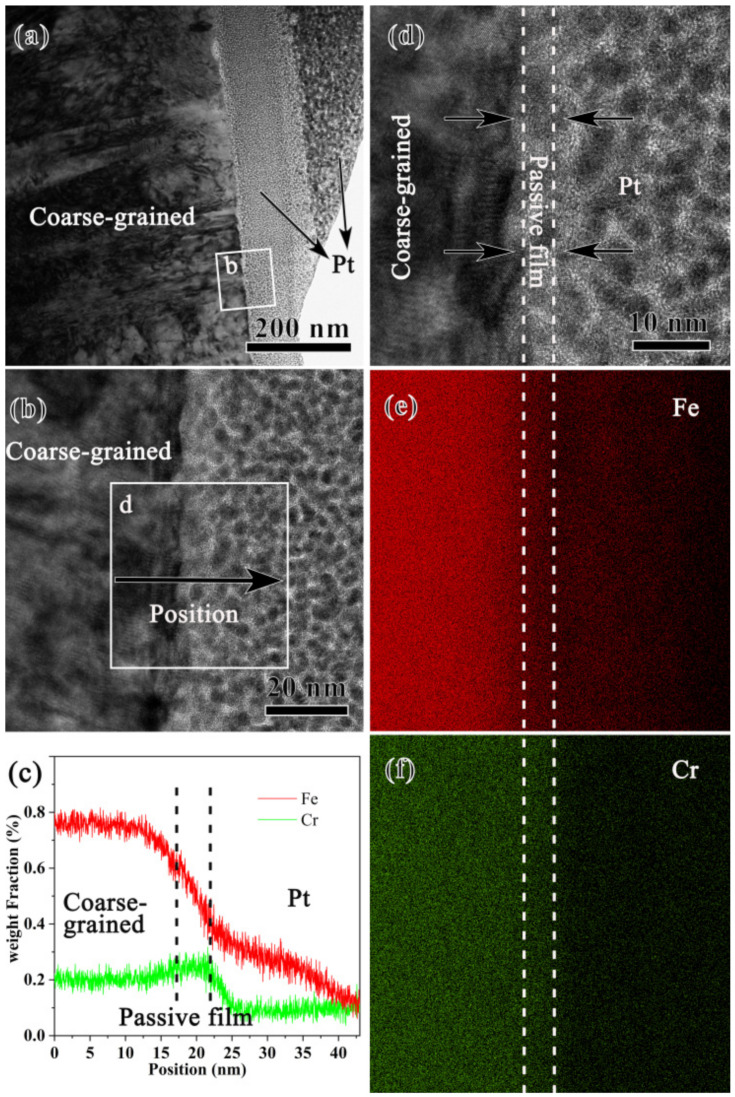
Bright-field TEM image (**a**), high-resolution TEM (HRTEM) image (**b**) of the squared area b in (**a**), EDS line scan (**c**) along the black arrow in (**b**), the HRTEM image (**d**) of the area d in (**b**), and EDS maps of Fe (**e**) and Cr (**f**) of the coarse-grained sample after pit corrosion in 0.6 mol/L NaCl aqueous solution.

**Figure 4 nanomaterials-13-00318-f004:**
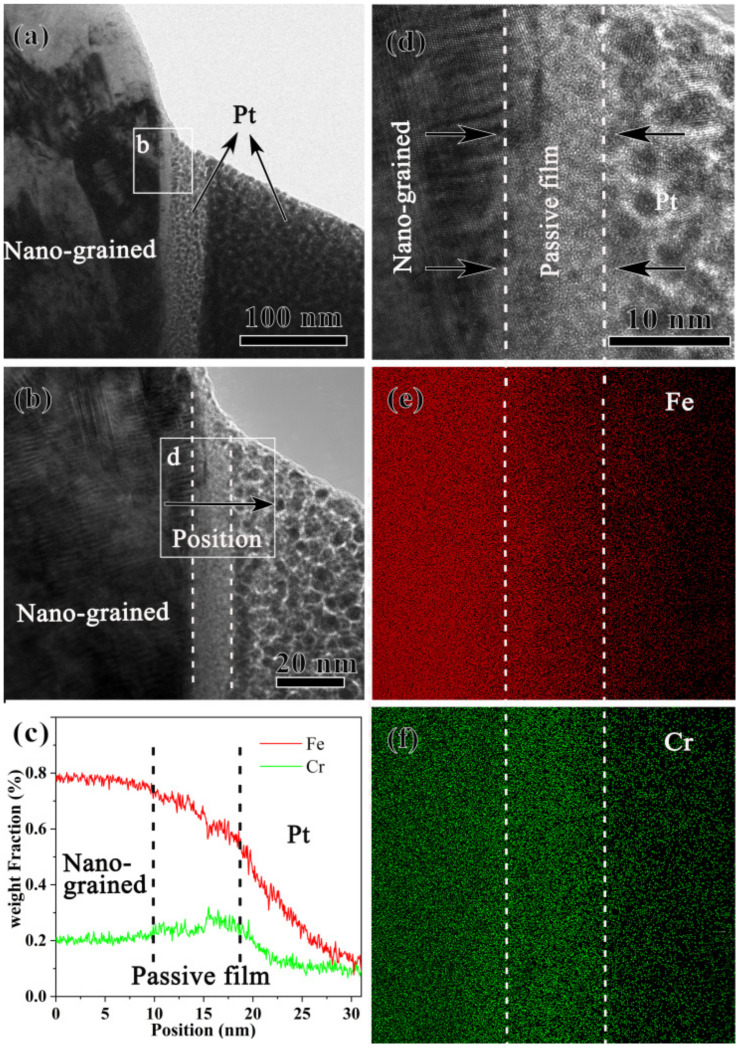
Bright field TEM image (**a**), high-resolution TEM image (**b**) of the area b in (**a**), EDS line scan (**c**) along the black arrow in (**b**), the high-resolution TEM image (**d**) of the area d in (**b**), and EDS maps of Fe (**e**) and Cr (**f**) of the nanograined sample after pit corrosion in 0.6 mol/L NaCl aqueous solution.

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
