# Peer review of "Enhanced Pitting Corrosion Resistance of Nanostructured AISI 304 Stainless Steel via Pipe Inner Surface Grinding Treatment"

_nanomaterials, 2023, doi:10.3390/nano13020318_

Round 1
Reviewer 1 Report
This article dealing with the formation of a single-phase nanostructured austenite on the surface of an AISI 304 stainless steel via a pipe inner surface grinding. This article looks very appealing, however, some revisions needs to be done before the publication.
1). Page 2, line 50. Please add a small paragraph depicting the gap between this investigation and the previous work.
2). Page 2, line 66. Could you please provide a photo of device for the PISG processing?
3). Page 2, line 74, How did you determine the parameters for the PISG processing?
4). Part 5, line 261. I guess this part needs to be rewritten. Some quantitive results need to be summarized in this part.
Author Response
请参阅附件。

Reviewer 2 Report
The subject of paper is scientifically interesting. However, in order to accept, the paper should be subjected to a careful revision. The detailed remarks are as follows:
- The literature survey presented in paper seems limited. Authors are requested to extend the state-of-the-art by consideration of the newest works within this scope (related to processing of stainless steels and precise machining). Among others, the following ones should be studied and included in a literature survey.
1. 10.1016/j.triboint.2020.106334
- At the end of Introduction please clearly present the novelty of the proposed method towards others in this area.
- In order to improve the readability of conclusions please formulate them in a form of a bullet points depicting the main findings both in the qualitative and quantitative way
Round 2
Reviewer 2 Report
Authors have revised their manuscript in accordance with all my remarks. Thus, the paper can be accepted for a publication in a current form.